# Durvalumab plus pazopanib combination in patients with advanced soft tissue sarcomas: a phase II trial

Hee Jin Cho[1,13], Kum-Hee Yun ®[2,13], Su-Jin Shin[3,13], Young Han Lee[4], Seung Hyun Kim ®[5], Wooyeol Baek ®[6], Yoon Dae Han[7], Sang Kyum Kim[8], Hyang Joo Ryu[8], Joohee Lee[4], Iksung Cho[9], Heounjeong Go ®[10], Jiwon Ko[10,11], Inkyung Jung ®[12], Min Kyung Jeon[2], Sun Young Rha ®[2] & Hyo Song Kim ®[2] ✉

We aimed to determine the activity of the anti-VEGF receptor tyrosine-kinase inhibitor, pazopanib, combined with the anti-PD-L1 inhibitor, durvalumab, in metastatic and/or recurrent soft tissue sarcoma (STS). In this single-arm phase 2 trial (NCT03798106), treatment consisted of pazopanib 800 mg orally once a day and durvalumab 1500 mg once every 3 weeks. Primary outcome was overall response rate (ORR) and secondary outcomes included progression-free survival (PFS), overall survival, disease control rate, immune-related response criteria, and safety. The ORR was 30.4% and the trial met the pre-specified endpoint. The median PFS was 7.7 months (95% confidence interval: 5.7–10.4). The common treatment-related adverse events of grades 3–4 included neutropenia (9 [19.1%]), elevated aspartate aminotransferase (7 [14.9%]), alanine aminotransferase (5 [10.6%]), and thrombocytopenia (4 [8.5%]). In a prespecified transcriptomic analysis, the B lineage signature was a significant key determinant of overall response ($P = 0.014$). In situ analysis also showed that tumours with high CD20$^+$ B cell infiltration and vessel density had a longer PFS ($P = 6.5 \times 10^{-4}$) than those with low B cell infiltration and vessel density, as well as better response (50% *vs* 12%, $P = 0.019$). CD20$^+$ B cell infiltration was identified as the only independent predictor of PFS via multivariate analysis. Durvalumab combined with pazopanib demonstrated promising efficacy in an unselected STS cohort, with a manageable toxicity profile.

Soft tissue sarcoma (STS) comprises a heterogeneous group of tumours with distinct clinical and molecular features and accounts for 1% and 15% of adult and paediatric tumours, respectively. Single agent doxorubicin is the preferred fist-line treatment for advanced STS, whereas anthracycline-based combinations (doxorubicin or epirubicin with ifosfamide) can be considered to relieve symptoms with significant tumors[1,2]. Although it did not show improved overall survival compared to that with doxorubicin monotherapy, the gemcitabine/docetaxel combination can be considered based on the GeDDiS trial[3]. Beyond the first-line setting, trabectedin, eribulin, pazopanib and gemcitabine/docetaxel regimens are available treatment options for selected subtypes[4–7]. However, these therapies still have modest efficacies, with objective response rates of 10–18% and a median progression-free survival (PFS) of 4 months, prompting the need for novel strategies.

Although monotherapy with an anti-PD-1 antibody and combination therapy with cytotoxic T-lymphocyte associated protein 4 (CTLA-4) antibodies have shown anti-tumour activity in advanced sarcomas, the response rate remains modest less than 20%[8,9]. Based on the main role of vascular endothelial growth factor (VEGF) in tumour

angiogenesis and immunosuppression[10], co-inhibition of VEGF and PD-1 signalling showed promising activity in melanoma, renal cell carcinoma, and some sarcoma subtypes[11,12]. A combination of VEGF and PD-1 blockade also showed promising preliminary activity and manageable toxicity for STS[13]. However, a third of the study investigated VEGF-dependent subtypes such as alveolar soft part sarcoma (ASPS), and the mechanism underlying the efficacy of these agents remaining elusive. Furthermore, the nivolumab/sunitinib combination had favourable activities, with 48% of 6-month PFS rate and 24 months of overall survival (OS), but they used lower dose of sunitinib (25 mg) due to high rate of dose interruption and toxicity[14]. Based on increasing evidence, to confirm a better efficacy and toxicity profile, further investigation with a combination of VEGF and PD-1/PD-L1 blockade for various subtypes of STS is needed. Further, to gain a better understanding of the determinants for response, it is necessary to establish biomarkers for identifying patients who would likely benefit from combination treatment.

In this work, we conducted an open-label, phase 2 study to determine the activity of the anti-VEGF receptor tyrosine-kinase inhibitor, pazopanib, combined with the anti-PD-L1 immune checkpoint inhibitor, durvalumab, in unselected advanced sarcomas. In addition, we conducted whole exome and transcriptomic sequencing with pre-treatment tissue biopsy to correlate clinical outcomes with molecular and genomic biomarkers to identify patients who would most likely benefit from the combination treatment.

## Results

### Clinical and pathological characteristics and treatment efficacy

Between April 2019 and October 2020, 48 participants were recruited, 47 (97.9%) of whom enrolled and received trial treatment (Supplementary Fig 1). Forty-six of the 47 patients were evaluable for response. The demographic and clinical characteristics of the patients are presented in Table 1.

As of the time of data cut-off (March 1, 2022), 6 remains on treatment with a median follow-up duration of 18.4 months. Of the 46 patients evaluable, 1 (2.2%) achieved complete response (CR), 13 (28.3%) achieved confirmed partial response (PR), and 27 (58.7%) had stable disease (SD), yielding an ORR of 30.4% and a DCR of 89.1% (Fig. 1A, B). The median time to PR was 1.7 months (range, 1.1–5.6), and the median duration of response was 8.3 months (range, 2.4–22.0). The most response occurred across various subtypes with ASPS, angiosarcoma (ANG), undifferentiated pleomorphic sarcoma (UPS), and desmoplastic small round cell tumour, while no response was reported in patients with leiomyosarcoma (LMS), dedifferentiated liposarcoma (DDLPS), and myxofibrosarcoma (Supplementary Table 1). Response assessment with RECIST was mostly concordant with irRC in which 2 patients with PR and PD according to RECIST were classified as having SD by irRC (Supplementary Table 2). PD-L1 expression (combined positive score ≥1) was observed in 50% of participants ($n = 23$), and it was not correlated with responses ($P = 0.58$).

Thirty-seven patients (80.4%) had progressive disease, and the median PFS was 7.7 months (95% CI, 5.7–10.4, Fig. 2A). The median OS was not reached with 1-year OS of 71.7% (Fig. 2B). No correlation was observed between the PD-L1 score and PFS ($P = 0.29$) or OS ($P = 0.052$) as shown in Supplementary Fig. 2.

### Safety

Among a total of 47 patients in the safety analysis, adverse events of any cause led to the discontinuation of durvalumab in 2 (4.3%) and dose reduction of pazopanib in 24 (51.1%) [600 mg in 16 (34.0%), 400 mg in 7 (14.9%), and 200 mg in 1 (2.1%) patients] (Fig. 2C). The mean durations of treatment with durvalumab and pazopanib were 7.1 and 7.6 months, respectively. The mean (±SD) dose intensities were 82.0% (±17.0) for pazopanib and 93.4% (± 8.4) for durvalumab. The median time to first dose reduction for pazopanib due to adverse events of any cause was 1.2 months (range, 0.7–8.7).

The common treatment-related adverse events of any grade included fatigue ($n = 20$, 42.6%), anorexia ($n = 17$, 36.2%), diarrhoea ($n = 17$, 36.2%), and elevated aspartate aminotransferase (AST) levels ($n = 14$, 29.8%) (Table 2). Treatment-related adverse events of grade 3 or 4 occurred in 19 (40.4%) patients, comprising neutropenia ($n = 9$, 19.1%), AST level elevation ($n = 7$, 14.9%), alanine aminotransferase

## Table 1 | Patient characteristics

| Variables | Total ($n = 47$) |
|---|---|
| **Age (Median, range)** | 51 (22–72) |
| **Gender** | |
| Male | 22 (46.8%) |
| Female | 25 (53.2%) |
| **ECOG PS** | |
| 0 | 23 (48.9%) |
| 1 | 21 (44.7%) |
| 2 | 3 (6.4%) |
| **Histologic variant** | |
| Leiomyosarcoma | 12 (25.5%) |
| Malignant peripheral nerve sheath tumor | 5 (10.6%) |
| Synovial sarcoma | 4 (8.5%) |
| Myxofibrosarcoma | 4 (8.5%) |
| Desmoplastic small round cell tumor | 4 (8.5%) |
| Undifferentiated pleomorphic sarcoma | 4 (8.5%) |
| Dedifferentiated liposarcoma[a] | 3 (6.4%) |
| Clear cell sarcoma | 2 (4.3%) |
| Endometrial stromal sarcoma | 2 (4.3%) |
| Alveolar soft part sarcoma | 2 (4.3%) |
| Angiosarcoma | 2 (4.3%) |
| Others[b] | 3 (6.4%) |
| **Primary site** | |
| Abdomen/retroperitoneum | 12 (25.5%) |
| Lower extremity | 10 (21.3%) |
| Head/neck | 10 (21.3%) |
| Genitourinary/gynecologic organ | 9 (19.1%) |
| Upper extremity | 4 (8.5%) |
| Thorax | 2 (4.3%) |
| **Previous resection** | |
| Yes | 41 (87.2%) |
| No | 6 (12.8%) |
| **Lines of systemic therapy** | |
| 1 | 40 (85.1%) |
| 2 | 7 (14.9%) |
| **Type of previous chemotherapy received ($n = 54$)** | |
| Doxorubicin/ifosfamide | 20 (37.0%) |
| Doxorubicin monotherapy | 15 (27.8%) |
| Ifosfamide combination (VIP/P6) | 7 (13.0%) |
| Gemcitabine/docetaxel | 6 (11.1%) |
| Others[c] | 4 (7.4%) |
| Eribulin | 2 (3.7%) |

[a]SCNAs revealed differences between histologies, 3 cases initially diagnosed as LMS ($n = 2$, YCC#9 and #40) and synovial sarcoma ($n = 1$, YCC#4) were revised as DDLPS based on molecular results with *CDK4* and *MDM2* co-amplification,
[b]Others: hemangioendothelioma, malignant glomus tumor, stromal sarcoma
[c]Others: paclitaxel ($n = 2$) and cyclophosphamide ($n = 2$)
*ECOG PS* Eastern Cooperative Oncology Group Performance Status

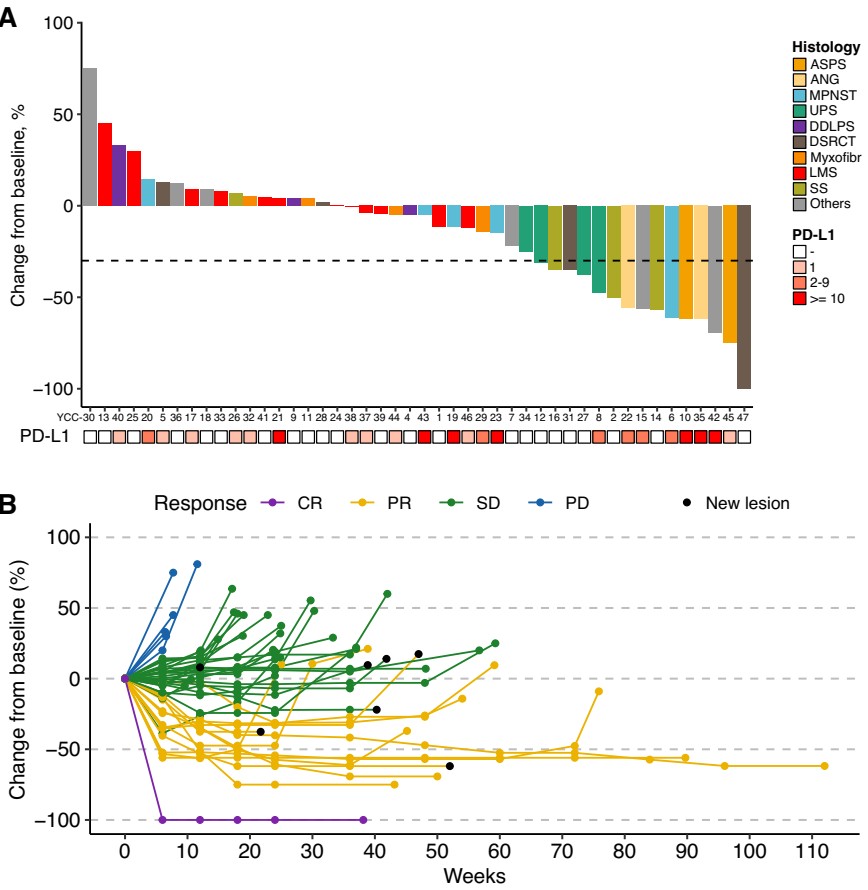

**Fig. 1 | Efficacy of durvalumab and pazopanib combination in the treatment of STS (*n* = 46). A** Waterfall plot of the maximum change in tumour size. From top to bottom, panels indicate: Waterfall plot representing the percentage of maximum tumour reduction as assessed according to RECIST 1.1 criteria, including histological subtype and PD-L1 expression. Others indicate clear cell sarcoma (*n* = 2), endometrial stromal sarcoma (*n* = 2), malignant glomus tumor (*n* = 1), and hemangioendothelioma (*n* = 1). **B** Representative spider plot illustrating changes in tumour burden from the baseline. ASPS Alveolar soft part sarcoma, ANG angiosarcoma, MPNST malignant peripheral nerve sheath tumour, UPS undifferentiated pleomorphic sarcoma, DDLPS dedifferentiated liposarcoma, DSRCT desmoplastic small round cell tumour, LMS leiomyosarcoma, SS synovial sarcoma, CR complete response, PR partial response, SD stable disease, PD progressive disease. Source data are provided as a Source Data file.

(ALT) level elevation (*n* = 5, 10.6%), and thrombocytopenia (*n* = 4, 8.5%).

**Genomic landscape and immune microenvironment signature exploration.** For the prespecified transcriptomic analysis, tumor biopsies were obtained from 33 (70.2%) of 47 patients prior to treatment, and 28 (84.8%) of those passed quality control with successful analyses for genomic alterations (Supplementary Data 1). The overall tumour mutation burden was low (median = 41) and mutational analysis revealed recurrently mutated genes, including *TP53* (*n* = 5, 17.9%), *ATRX* (*n* = 3, 10.7%), and *NF1* (n = 3, 10.7%), none of which were correlated with the response to treatment (Fig. 3A). In addition, amplification and homozygous deletions of any genes were not associated with the response. Although *CDKN2A* alterations combining both heterozygous and homozygous deletions were significantly associated with response (*P* = 0.0028), presence of homozygous deletion was not statistically significant (P = 0.53). We evaluated the association between treatment responses and tumour mutation burden (TMB), neoantigen burden, and human leukocyte antigen (HLA) loss of heterozygosity (LOH), which have been reported to be potential predictive biomarkers for immunotherapy in other tumour types[15–17]. Among those, only TMB was statistically related to the treatment response in our sarcoma cohort (Supplementary Fig 3).

Based on transcriptomic data (Supplementary Data 2), we examined the association between known tumour microenvironment

(TME)-related genes and treatment response across histological subtypes (Fig. 3B). Using MCP-counter based on the expression of TME-related genes[18], we estimated the scores of TME composite cells such as T cells, CD8+ T cells, natural killer (NK) cells, endothelial cells, and cytotoxic lymphocytes, and among them, B lineage signature was a significant key determinant for treatment response (*P* = 0.014, Fig. 3C). In addition, through differentially expressed gene (DEG) analysis performed by DESeq2[19], the marker genes of MCP-counter were compared between responders and non-responders to the treatment, and three (CD19, CD79A, and MS4A1) of eight B lineage markers were significantly up-regulated in responders, which was the highest proportion of significantly up-regulated genes among the TME cell types (Supplementary Fig. 4). Regarding the TME signatures and the mRNA expression levels of immune-checkpoint-related genes encoding PD1, PDL1, CTLA4, and LAG3, the differences between responders and non-responders were not significant (Fig. 3C and Supplementary Fig 5). Upon application of the Sarcoma Immune Classes (SIC)[20] classification predicted by the signature scores of the microenvironment cell populations (MCP)-counter except fibroblast, a B lineage-high signature (a hallmark of an immune-high E class) correlated with an improved PFS, unlike class B (immune-low). Four of the five patients (80%) who fell into class E were also responsive to the treatment (Supplementary Fig 6), consistent with the previous pembrolizumab study[20]. Regarding the SIC E class (*n* = 5), DSCRT (*n* = 1) had CR, SS (*n* = 2) and UPS (*n* = 1) had PR, whereas LMS (*n* = 1) had SD, respectively.

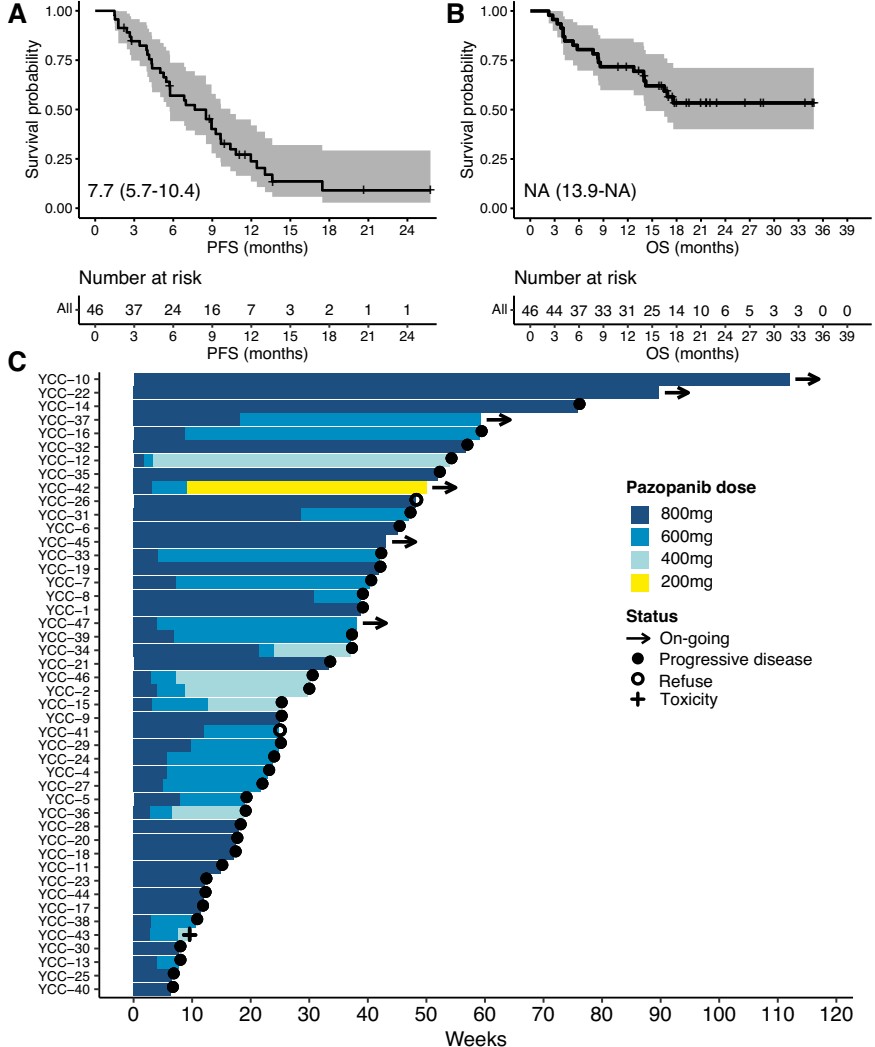

**Fig. 2 | Survival analyses of durvalumab and pazopanib combination (n = 46).** Kaplan–Meier curves associated with PFS (**A**) and OS (**B**) for all patients. **C** Swimmer plot. Each lane represents a single patient's data. X-axis represents the duration of treatment for each patient. PFS progression-free survival, OS overall survival. Source data are provided as a Source Data file.

## Multiplex immunohistochemistry validation for distinct tumour microenvironment

We validated immune cell and vessel densities in 39 primary tumour samples via quantitative immunohistochemistry and Opal multiplexed immunofluorescence staining in accordance with treatment efficacy (Supplementary Fig 7). Of those, CD3$^+$ T cell ($P = 7.01 \times 10^{-3}$), CD8$^+$ T cell ($P = 8.82 \times 10^{-4}$), and CD20$^+$ B cell ($P = 0.042$) infiltration, as well as CD31$^+$ vessel density ($P = 5.18 \times 10^{-3}$) showed significantly higher expression among responders (Supplementary Fig. 8). Similarly, when the analyses were performed for PFS for high (≥median) and low (< median) densities, CD20$^+$ B cell infiltration ($P = 0.0014$) and vessel density ($P = 0.023$) were significantly associated with improved PFS (Fig. 4A, B), whereas CD3$^+$ T cell, CD8$^+$ T cell, and PD1$^+$ cell infiltration were not (Supplementary Fig 9). Therefore, tumours with high CD20$^+$ B cell infiltration and vessel density had longer PFS than those with low CD20$^+$ B cell infiltration and vessel density ($P = 6.5 \times 10^{-4}$), as well as a higher response rate (50% *vs* 12%, Fisher's exact test $P = 0.019$, Fig. 4C). Finally, in the univariate Cox regression analysis including histology, SIC, clinical factors (age, gender, primary sites, and stage), immune and vessel densities, high CD20$^+$ B cell infiltration ($P < 0.01$), and high vessel density ($P = 0.03$) were associated with improved PFS. Of these, CD20$^+$ B cell infiltration was identified as the only independent

predictor of PFS in multivariate analysis (Supplementary Table 3). In addition, B lineage scores from the MCP-counter were significantly correlated with CD20+ density by immunofluorescence ($P = 0.0087$, R = 0.51 by Pearson, Supplementary Fig 10).

## Discussion

In this prospective clinical trial, we investigated the efficacy and safety of combined VEGF and PD-L1 blockade in different subtypes of STS. This study also evaluated the relationship between clinical outcomes and combined VEGF and PD-L1 blockade via integrative molecular analysis. Based on multiple modalities for STS TME, we found that high B lineage and vessel density exhibited the highest response and improved PFS in the presence of combined PD-L1 blockade and VEGF inhibition.

Our results obtained by combining VEGF and PD-L1 inhibition are promising. Despite recent improvements in STS treatment, drug activity remains moderate. Particularly, beyond the first-line treatment, an objective response rate of <16% and PFS of 2–4 months are reported[4–6]. Although cross-study comparisons are suppositional, the ORR obtained in our study was favourable compared to those among patients treated with pembrolizumab (15%), pazopanib (6%), or axitinib (4% for LMS) monotherapy[7,9,21] or nivolumab and ipilimumab in

**Table 2 | Incidence of all treatment-related adverse events (n = 47)**

| Event | Grade (n, %) | | | |
|---|---|---|---|---|
| | Grade 1, 2 | Grade 3 | Grade 4 | All |
| Fatigue | 18 (38.3%) | 2 (4.3%) | .. | 20 (42.6%) |
| Anorexia | 15 (31.9%) | 2 (4.3%) | .. | 17 (36.2%) |
| Diarrhea | 15 (31.9%) | 2 (4.3%) | .. | 17 (36.2%) |
| AST increased | 7 (14.9%) | 6 (12.8%) | 1 (2.1%) | 14 (29.8%) |
| Neutropenia | 3 (6.4%) | 8 (17.0%) | 1 (2.1%) | 12 (25.5%) |
| ALT increased | 7 (14.9%) | 4 (8.5%) | 1 (2.1%) | 12 (25.5%) |
| Hypothyroidism | 10 (21.3%) | .. | .. | 10 (21.3%) |
| Pyrexia | 9 (19.1%) | 1 (2.1%) | .. | 10 (21.3%) |
| Thrombocytopenia | 4 (8.5%) | 4 (8.5%) | .. | 8 (17.0%) |
| Nausea | 7 (14.9%) | .. | .. | 7 (14.9%) |
| Hypertension | 7 (14.9%) | .. | .. | 7 (14.9%) |
| Myalgia | 6 (12.8%) | | | 6 (12.8%) |
| Hand-foot syndrome | 4 (8.5%) | 2 (4.3%) | .. | 6 (12.8%) |
| Lipase increased | 3 (6.4%) | 2 (4.3%) | .. | 5 (10.6%) |
| Anemia | 3 (6.4%) | 1 (2.1%) | .. | 4 (8.5%) |
| Proteinuria | 4 (8.5%) | .. | .. | 4 (8.5%) |
| Oral mucositis | 4 (8.5%) | .. | .. | 4 (8.5%) |
| Pruritis | 3 (6.4%) | .. | .. | 3 (6.4%) |
| Constipation | 3 (6.4%) | .. | .. | 3 (6.4%) |
| Headache | 3 (6.4%) | .. | .. | 3 (6.4%) |
| Asthenia | 2 (4.3%) | .. | .. | 2 (4.3%) |
| Vomiting | 2 (4.3%) | .. | .. | 2 (4.3%) |
| Pneumonitis | 2 (4.3%) | .. | .. | 2 (4.3%) |
| ALP increased | 1 (2.1%) | .. | 1 (2.1%) | 2 (4.3%) |
| Amylase increased | 2 (4.3%) | .. | .. | 2 (4.3%) |
| Hyperbilirubinemia | 2 (4.3%) | .. | .. | 2 (4.3%) |
| Hyperthyroidism | 1 (2.1%) | .. | .. | 1 (2.1%) |
| Adrenal insufficiency | 1 (2.1%) | .. | .. | 1 (2.1%) |
| Creatinine elevated | .. | 1 (2.1%) | .. | 1 (2.1%) |
| Cardiac disorder | 1 (2.1%) | .. | .. | 1 (2.1%) |
| Arthritis | 1 (2.1%) | .. | .. | 1 (2.1%) |

*AST* Aspartate aminotransferase, *ALP* Alanine aminotransferase

combination (16%)[8]. Furthermore, the response to durvalumab and pazopanib in combination, as reported in this study, was comparable to that obtained with axitinib and pembrolizumab in combination[13]. In previous studies, various response rates were noted based on the proportion of ASPS (36% in the axitinib/pembrolizumab and 7% in sunitinib/nivolumab)[13,14]. In addition, the indolent nature of ASPS may influence PFS and OS. Therefore, although it is not statistically significant in multivariate analyses, we cautiously investigated the subgroup efficacy to provide further informative signals for clinicians. The non-ASPS cohort in our study achieved significantly better response (ORR: 27.3% vs 9.5%) and PFS (7.0 vs 3 months) than those obtained with the axitinib–pembrolizumab combination. Consequently, the ASPS as well as non-ASPS cohorts in our study showed favourable efficacies. This finding should be considered for hypothesis-generating purposes.

Possible important issues to note in our study are as follows. From a cautiously speculative standpoint, the pazopanib dose intensity in our study could explain the favourable outcome. In the study of axitinib and pembrolizumab, a dose escalation strategy (initiation of a lower dose of axitinib and escalation thereafter as 6, 7, 8, and 10 mg BID) was incorporated; consequently, only a 16% rate of intra-patient

dose escalation was established. On the other hand, we used a dose de-escalation strategy (initiation of a standard dose of pazopanib, 800 mg QD, and reduction thereafter). Consequently, toxicities of grade 1–2, such as fatigue, anorexia, and diarrhoea, were manageable, and the rate of treatment-related adverse events of grades 3–4 were consistent (approximately 40%) with that obtained with axitinib and pembrolizumab treatment. Interestingly, much higher grade 3–4 AST and/or ALT elevation occurred in our study than that observed with axitinib and pembrolizumab combination (approximately 15% vs 6%)[13]. This severe hepatotoxicity occurred within 1.5 months and resolved with temporary stoppage and/or short-term steroid use (Supplementary Fig 11). Eventually, only 2 patients with combined haematologic and liver toxicities permanently discontinued durvalumab. Considering that pazopanib yielded better efficacy with higher dose[22], we suggest a treatment strategy maintaining the standard dose of pazopanib to achieve better efficacy with tolerable toxicities in future trials. Simultaneously, further studies are also required for valuable biomarker exploration to identify patients who would benefit from combination treatment.

Despite the promising efficacy, little is known about the mechanism underlying the action of ICI and angiogenesis inhibitors in combination. Only limited numbers of studies have demonstrated the prognostic importance of the immune microenvironment and druggable markers in sarcoma with heterogenous significance[23–26]; Sorbye et al. reported a significant association of CD20 + B peritumoral lymphocytes with disease free survival[25] and PD-L1 or NK cells were also reported as significant prognostic factors with multiple STS[23,25]. However, most of these studies were performed with immunologic endpoints and only increased baseline plasma angiogenic activity showed such a correlation with treatment efficacy[13]. While TMB status is not useful predictor to ICI in sarcoma because neoantigen levels are not correlated with CD8 T cells[27], ours demonstrated marginal statistical significance and further study may be needed to clarify this finding. Regarding the tertiary lymphoid structure (TLS), only 3 cases were TLS + in our study (PR, n = 2; SD, n = 1). Finally, for the ICI and angiogenesis combination, only two of 14 responders could be detected via TLS positivity. Therefore, considering the controversial issues needing further optimization originating from the heterogeneity (sample and tumor types)[28], we focused on exploring clinically applicable and reliable biomarkers. In accordance with previous results[20], transcriptomic and multiplex IHC analyses in our cohort showed that B-cell lineage significantly correlated with PFS and response. Regarding the tumour vessel, a significant relationship between high vascular density or CD31 with better response or longer survival in patients treated with sunitinib were reported[29–31]. Therefore, to elucidate further determinants for the effects of the ICI and angiogenesis inhibitor combination, we performed an exploratory analysis with immune cells as well as microvessel density in archival formalin-fixed paraffin-embedded tissue samples[32,33]. Although microvessel density was only significant in univariate analysis, considering the importance of the immunomodulatory functions of the tumor vasculature, combining angiogenesis and immune-related biomarkers may be worth exploring. The correlation between B-cell infiltration or vessel density and response to the ICI and angiogenesis inhibitor combination should be validated in a clinical trial comprising a larger set of patients.

Histologic diversity with different pathologic and clinical features may result in limited success in developing checkpoint blockade in unselected populations[9,34]. There is strong evidence that UPS, ASPS, and ANG are associated with immunologic features of high levels of tumour-infiltrated immune cells accompanied by clinical efficacy with immunotherapy[8,9,35]. Given the dramatic response as well as high expression of immune and vessel densities demonstrated in UPS, ASPS, and ANG (Supplementary Fig 12), PD-L1 blockade and angiogenesis inhibitor combination can be considered as a preferred option. Although our results are promising, benefit as first-line treatment must

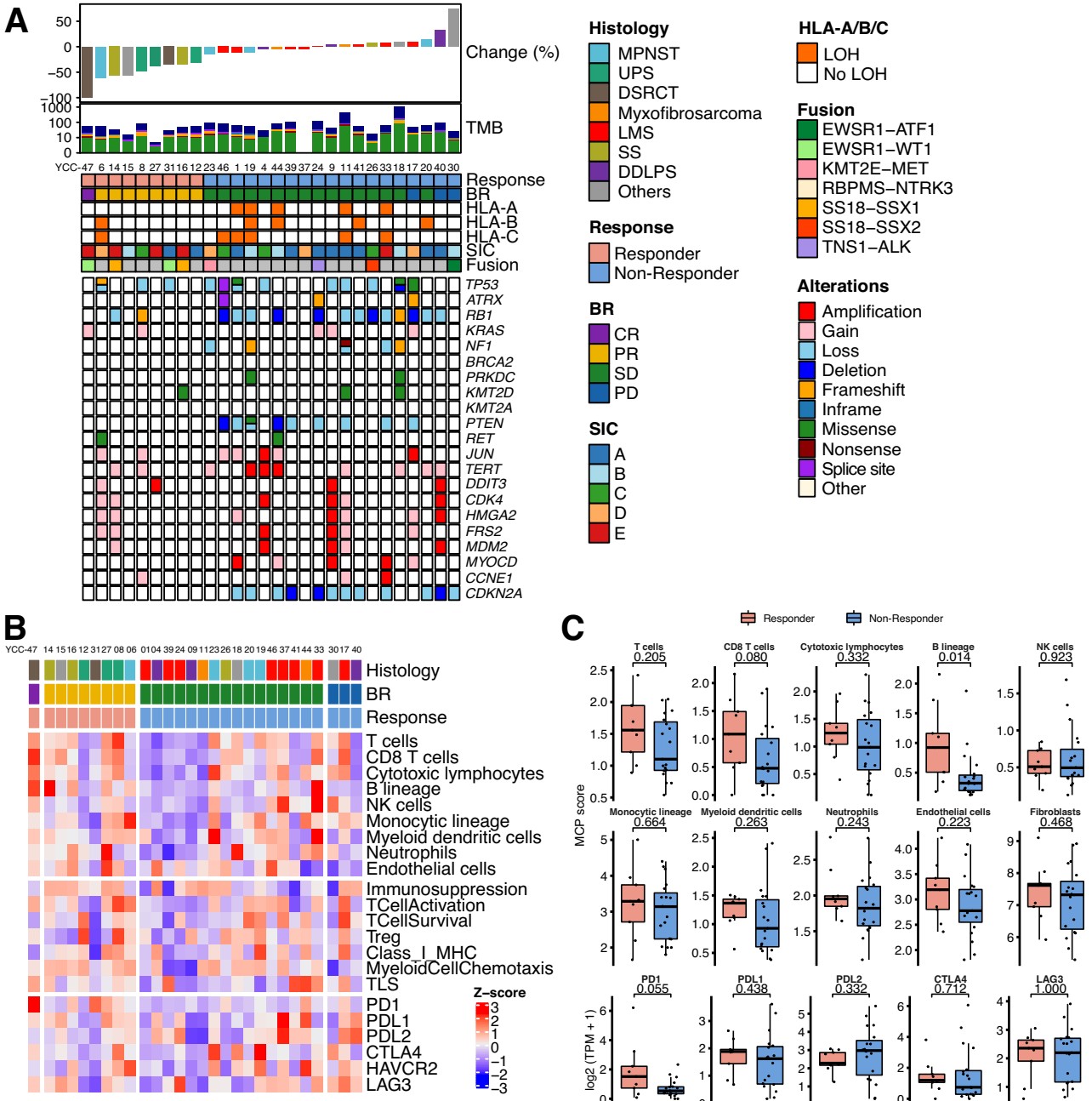

**Fig. 3 | Molecular landscape and response to durvalumab and pazopanib combination treatment (n = 28). A** Integrated plot of clinical and molecular features with whole-exome sequencing results From top to bottom, panels indicate: Waterfall plot representing the percentage of maximum tumour reduction as assessed according to RECIST 1.1 criteria; the number of somatic mutations; clinical characteristics including best response, histological subtype, and landscape of alterations (LOH of HLA genes, gene fusions, mutations, and somatic copy number alterations). **B** Transcriptomic correlates of clinical response to durvalumab and pazopanib combination treatment. Heat maps describing tumour microenvironment cell infiltration estimated by MCP-counter. From top to bottom, heat maps indicate: MCP-counter scores of immune and stromal cells; single sample gene set enrichment analysis (ssGSEA) scores for immune-associated gene signatures; and gene expression levels for immune-checkpoint genes. The colour scale indicates Z-normalised values of each gene signature for gene expression across samples. The colour bars above the heatmap indicate the tumor histology (top), best response (middle; CR, purple; PR, yellow; SD, green; PD, blue), and responders (pink) and non-responders (blue) to the pazopanib-durvalumab combination

(bottom). **C** Comparison of TME and immune-checkpoint genes between responders (n = 9) and non-responders (n = 19). Composition of the TME estimated using MCP-counter scores were compared between responders and non-responders (top). Expression levels of immune-checkpoint genes were compared between responders and non-responders (bottom). Center lines, upper and lower bounds of boxplots indicate the median, 25th, and 75th quantile, respectively. The whiskers of boxplots indicate 1.5 times of the interquartile range. *P*-value was derived from the two-sided Wilcoxon rank-sum test and not adjusted for multiple tests. TME tumour microenvironment, TMB tumour mutation burden, HLA human leukocyte antigen, MPNST malignant peripheral nerve sheath tumour, UPS undifferentiated pleomorphic sarcoma, DSRCT desmoplastic small round cell tumour, LMS leiomyosarcoma, SS synovial sarcoma, DDLPS dedifferentiated liposarcoma, BR best response, CR complete response, PR partial response, SD stable disease, PD progressive disease, LOH loss of heterozygosity, Treg regulatory T cells, Class_I_MHC major histocompatibility complex class I, TLS tertiary lymphoid structures, HAVCR2 Hepatitis A Virus Cellular Receptor 2, LAG3 Lymphocyte-Activation Gene 3. Source data are provided as a Source Data file.

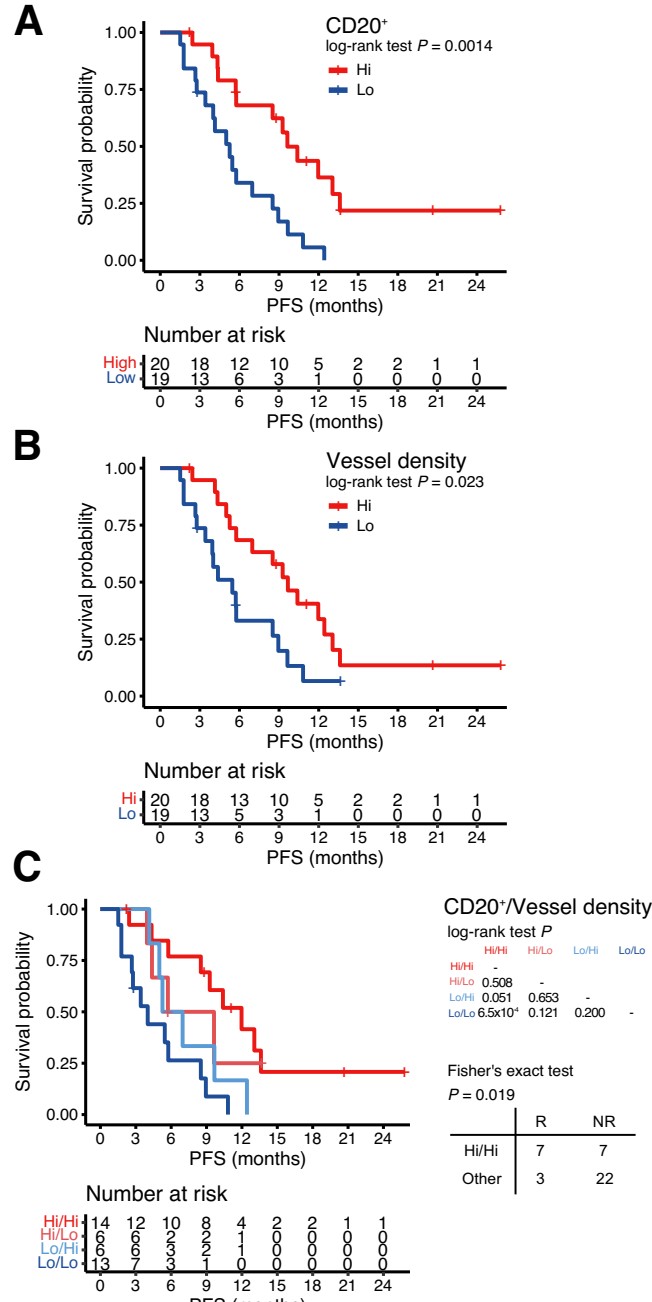

**Fig. 4 | Distinct quantification of the immune microenvironment by multiple IHC analysis (n = 39).** PFS of patients according to CD20+ B cell infiltration (**A**) and vessel density (**B**). **C** PFS and response of patients based on CD20+ cell infiltration and vessel density. The right-bottom panel indicates the number of samples grouped by response to the treatment and CD20+ B cell infiltration and vessel density, and the association of response and CD20+/vessel density group was evaluated via two-sided fisher's exact test. P < 0.05 was considered statistically significant. Hi indicates patients with a density higher than or equal to the median, and Lo indicates patients with a density lower than the median. IHC immunohistochemistry, PFS progression free survival, R responder, NR non-responder. Source data are provided as a Source Data file.

be confirmed with larger randomized trials. On the contrary, LMS and DDLPS present molecular challenges. Most cases of LMS are known to have low immune infiltration[20], and ICI as monotherapy or in combination with axitinib did not show any clinical activity[8,13,36]. A heterogenous spectrum of immune profiles was also found in DDLPS[20,37], in which three distinct clusters were categorised with different immune

signatures[23]. Despite the small sample size, cases of LMS and DDLPS in our study showed modest efficacy, and the B cell expression and vessel density were also generally low as shown in Supplementary Fig 12. Based on our result in line with a prior clinical study with favourable efficacy of CDK4/6 inhibitor[38], a rational treatment strategy will require a molecular understanding of specific histological types.

A limitation of our study was that it was not designed to directly compare clinical safety and activity with that of pazopanib monotherapy. In addition, despite the comprehensive analysis, a majority of responders, including those with ASPS and ANG, were not included in the analysis of transcriptomic sequencing because of the inaccessibility for tumour extraction. Despite these limitations and challenges, our study made several important contributions. Unlike a previous study that mainly recruited patients with VEGF-dependent subtypes[13], we enrolled patients with heterogeneous subtypes collectively, thereby minimising selection bias of efficacy analyses. We also simultaneously assessed clinical efficacy in addition to matched genomic profiling using an Opal multiplexed staining platform. Recently, we have launched randomised clinical trials comparing the activity of PD-1 blockade and pazopanib in combination versus pazopanib as monotherapy (NCT05679921). Therefore, further investigations of VEGF and PD-1 blockade in combination with correlative markers might help elucidate the accurate efficacy and mechanisms underlying clinical responses.

In conclusion, the combination of pazopanib and PD-L1 blockade showed acceptable toxicity and promising efficacy in patients with previously treated advanced STS. This therapeutic approach of combining VEGF and PD-L1 blockade may prove a silver lining in the future care of patients with STS. Our study also identified the potential role of B-cell infiltration as a valuable clinical decision-making tool in the prognostication of this heterogeneous tumour, although further investigations are needed.

## Methods
### Patient selection and study procedure
Patients enrolled in this study had histologically confirmed metastatic and/or recurrent STS as per the following inclusion criteria: (1) aged at least 19 years, (2) previous failure of one or two lines of chemotherapy, (3) at least one measurable lesion according to Response Evaluation Criteria In Solid Tumours (RECIST) 1.1[39], (4) Eastern Cooperative Oncology Group performance status of 0 or 1, and (5) adequate organ function per the protocol. All patients were naive to anti-PD-1, anti-PD-L1 antibodies, and/or pazopanib treatment. The trial protocol was approved by the institutional review board of participating centre (Yonsei Cancer Center, Yonsei University College of Medicine, Seoul, Republic of Korea) and registered at ClinicalTrials.gov (NCT03798106) on January 09, 2019. On July 01, 2019, the sample size was increased from 37 to 46 as power increased. The first patient was enrolled on April 10, 2019, and the last was recruited on October 06, 2020. All patients provided written informed consent before enrolment in accordance with the Declaration of Helsinki and the Guidelines for Good Clinical Practice.

This prospective trial was designed as a single-arm, phase 2 study at an academic cancer centre, and participants were recruited from all over the country. The treatment consisted of pazopanib 800 mg administered orally, once a day, continuously, and durvalumab 1500 mg via 60 min intravenous infusion once every 3 weeks until documented disease progression or unacceptable toxicity (Supplementary Fig 13). Dose reduction by 200 mg to a lowest dose of 400 mg were allowed based on the protocol. Durvalumab dose reductions were not permitted; however, subsequent infusions could be omitted in response to persisting toxic effects. Patients underwent imaging with contrast-enhanced images of all sites of disease at baseline and every 6 weeks for the first 24 weeks, and then every 12 weeks thereafter. Tumour responses were determined by investigators with an

independent radiologist by RECIST 1.1, with responses confirmed by a second scan at least 4 weeks after criteria for objective response were met. If progressive disease was confirmed at the subsequent 4 weeks assessment, the date of the initial progressive disease was used for analyses and the patient stopped study treatment. Adverse events were graded according to the National Cancer Institute Common Terminology Criteria for Adverse Events version 4.03.

## Correlative science
Optional biopsies of tumor site were performed before treatment and whole blood was obtained from patients for DNA germ line control. Tumours with estimated content ≥40% after pathological review were subjected to tumour DNA and RNA extraction from freshly obtained tissues using the DNeasy Blood & Tissue Kits (Qiagen, Hilden, Germany), RNase A (cat. #19101; Qiagen) and RNeasy Mini Kit (cat. #74106; Qiagen), respectively, according to the manufacturer's instructions. Germline genomic DNA was extracted from whole-blood samples. Analysis pipeline details for whole exome and RNA sequencing are available online in the supplementary information. Genomic information of 17 cases of this study was previously reported[40].

The tyramide signal amplification (TSA)-based Opal method was used for immunofluorescence staining. All multiplexed staining procedures were performed using the Opal 7 Immunology Discovery Kit (OP7DS2001KT; Akoya Biosciences, MA, USA). Primary antibodies directed against the following complexes were used: CD3 (1:50, UCHT1; Thermo Fisher Scientific, MA, USA), CD8 (1:200, 108M-96; Cell Marque Corp., CA, USA), CD20 (1:200, M0755; Dako Products, CA, USA), and PD1 (1:100, NAT105; Cell Marque Corp.). Immunohistochemical (IHC) staining for CD31 was performed using the Benchmark® automatic immunostaining device (Roche Tissue Diagnostics, Tucson, USA), according to the manufacturer's instructions. PD-L1 expression was assessed using the clone 22C3 (Agilent Technologies, CA, USA/ Dako Products, CA, USA) and the Dako Link 48 system (Agilent). All tissue specimens was reviewed and analyzed by independent pathologists (S.S. and H.G.). Manual annotation was performed to identify tumor regions of interest in all slides stained with hematoxylin-eosin (H&E). The density of specific cell types expressing CD3, CD8, CD20, or PD-1 was quantified by counting cells per mm$^2$. For CD31, positive density was determined as the proportion of positive pixels (including weak, moderate, and strong) relative to the total pixels within the regions of interest. These analyses were conducted on baseline tumor samples with accessible tumor material. Further information can be found in the supplementary materials.

## Study design and statistical analysis
With optimal two-stage design, the design had a one-sided type I error of 5% and a power of 90% to detect a difference in objective response between 5% and 20%. Treatment was considered successful if 5 or more of the 41 evaluable patients had a partial response or better. Allowing for a follow-up loss rate of 10%, the total sample size was expected to be 46 patients.

The primary endpoint of the trial was the overall response rate (ORR) as per RECIST 1.1. The secondary endpoints included the disease control rate (DCR), PFS, overall survival (OS), safety profile, Immune-Related Response Criteria (irRC), and exploratory biomarker analysis results. The ORR was calculated as the percentage of patients experiencing a confirmed CR or PR, and the DCR was calculated as the sum of the CR, PR, and stable disease rate. The PFS was defined as the time from the start of treatment to the date of disease progression or death resulting from any cause. The OS was measured as the time from the start of treatment to the date of death from any cause. Safety analysis included all patients who received at least one dose of treatment. Patients who received at least one dose of treatment and had undergone at least one disease assessment were included in the efficacy

analysis. Statistical associations between continuous and categorical variables were evaluated using Wilcoxon rank-sum statistics (two-sided). Genomic correlates of response to treatment were evaluated by Fisher's exact test. Survival was plotted using Kaplan–Meier curves and compared using the log–rank test. All statistical analyses were performed using R version 4.0.5 (The R Development Core Team, Vienna, Austria).

## Reporting summary
Further information on research design is available in the Nature Portfolio Reporting Summary linked to this article.

## Data availability
The list of mutations and gene count data are presented in Supplementary Data 1 and 2. WES and RNA sequencing data generated in this study are available upon reasonable request due to privacy laws related to the patients' content for data sharing and the data should be used for research purposes only. The raw sequencing data have been deposited in the EGA database under the accession study number EGAS50000000082. Data access can be granted via the EGA with the completion of an institute data transfer agreement (EGA Data Access Committee: EGAC50000000046), and data will be available for a defined time period once access has been granted. Additional de-identified clinical data can be made available upon request from the corresponding author (hyosong77@yuhs.ac). In case of publication(s) from the requester, the data will be available for 2 years from the date of last publication. If there is no use of the data for a period of 2 years, the requester must delete the data. The study protocol is available as a Supplementary Note in the Supplementary Information file. The remaining data are available within the Article, Supplementary Information, or Source Data file. Source data are provided with this paper.

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

## Acknowledgements
This work was also supported by the National Research Foundation of Korea (NRF) grant funded by the Korean government (MSIT) (2021R1A2C1094530 and RS-2023-00209741) and a faculty research grant of Yonsei University College of Medicine (6-2019-0187). The study drug durvalumab (IMFINZI®) was kindly provided by AstraZeneca Pharm. The funder had no role in study design, data collection, data analysis, data interpretation, or in the writing of this report.

## Author contributions
H.S.K. contributed to the conception and design of the study and obtained research funding. H.S.K., Y.H.L., S.H.K., W.B., Y.D.H., J.L., I.C. and S.Y.R. treated study patients. S.S., and J.K. conducted all the pathological review and analyses. S.K.K., H.G., K.H.Y., and H.J.R. processed the clinical samples. H.J.C., I.J, and M.K.J. conducted genomic and transcriptomic data analysis and statistical tests. H.S.K., H.J.C., K.H.Y., and S.S. wrote the manuscript. All authors discussed and approved the final manuscript.

## Competing interests
H.S.K. received grant/research support from MSD, Eli Lilly and Company, Ono Pharmaceutical Company, Medpacto Pharmaceutical, and Boryung Pharmaceutical Company outside the submitted work. S.Y.R. received grant/research support from MSD, Celltrion, Boehringer-Ingelheim, Eli Lilly and Company, Taiho, Bristol-Myers Squibb, ASLAN, and Incyte; consultation fees for Daiichi Sankyo, MSD, Eli Lilly, Bristol-Myers Squibb, and Eisail; and served on a speaker's bureau for Eli Lilly, Bristol-Myers Squibb, and MSD outside the submitted work. All remaining authors declare no competing interests.

## Additional information

[1]Department of Biomedical Convergence Science and Technology, CMRI, Kyungpook National University, Daegu, Republic of Korea. [2]Division of Medical Oncology, Department of Internal Medicine, Yonsei University College of Medicine, Seoul, Republic of Korea. [3]Department of Pathology, Gangnam Severance Hospital, Yonsei University College of Medicine, Seoul, Republic of Korea. [4]Department of Radiology, Yonsei University College of Medicine, Seoul, Republic of Korea. [5]Department of Orthopaedic Surgery, Yonsei University College of Medicine, Seoul, Republic of Korea. [6]Department of Plastic Surgery, Yonsei University College of Medicine, Seoul, Republic of Korea. [7]Department of Surgery, Yonsei University College of Medicine, Seoul, Republic of Korea. [8]Department of Pathology, Severance Hospital, Yonsei University College of Medicine, Seoul, Republic of Korea. [9]Division of Cardiology, Severance Cardiovascular Hospital, Yonsei University College of Medicine, Seoul, Korea. [10]Department of Pathology, Asan Medical Center, University of Ulsan College of Medicine, Seoul, Republic of Korea. [11]Asan Institute for Life Sciences, Asan Medical Center, Seoul, Republic of Korea. [12]Division of Biostatistics, Department of Biomedical Systems Informatics, Yonsei University College of Medicine, Seoul, Republic of Korea. [13]These authors contributed equally: Hee Jin Cho, Kum-Hee Yun, Su-Jin Shin. ✉e-mail: hyosong77@yuhs.ac

