## [Peer Review File · Nature Communications]

Durvalumab plus Pazopanib Combination in Patients with Advanced Soft Tissue Sarcomas: a phase II trialREVIEWER COMMENTS

Reviewer #1 (Remarks to the Author): with expertise in sarcoma (immuno)therapy

Nice to see validation of SICE/B cell lineage and correlation w ORR

Specifically interesting to see CD20 B cell / vessel density

Overall a nice manuscript, however, the background and info on sarcoma as a disease, and approach to care is not correct in both introduction/conclusion and should be corrected. Also this is a small patient cohort, and the results seem to be overstated.

Abstract

-Highlight total TRAE grade $\frac{1}{2}$ and $\frac{3}{4}$ rate

Introduction

-Based on the Geddis trial demonstrated that dox = gem/doce, and considered first line option as well. AIM has never demonstrated OS benefit and really reserved for pts w high tumor burden/symptoms.

-Gem doce ORR is actually 18%, not <10% as above

-2nd paragraph, should also highlight the nivo/sutent trial (immunosarc) JITC 2020 Broto et al.

- Overall helpful to include prior IO + TKI studies/outcomes

-Would fix intro to reflect appropriate treatment strategies/data in STS

Results

-Rationale in comparing ang/asps cohorts vs other, was this a predefined endpt? Would caution this analysis

-4 pts that were SICE, what is their histology

-Comment which were in SICE, but did not respond

Methods

-How was vessel density performed?

-Method section is very scant, unclear how analyses were actually performed; not clear how to reproduce

Conclusion

-Dox is not the only first line, refer to comment above

-Would caution analysis w nonASP/Ang vs other, there is no clear precedence to do this, instead just highlight the ORR, and which histologies stood out

-Biomarkers of response in sarcoma that have correlated w efficacy include: TIL, PD1 expression - this should be noted

-Would explain significance of vessel density as this is novel and how pazopanib may contribute to this density (or not)

-Difficult to conclude that durvalumab + pazo should be front line; there are <10 pts treated on this trial w these histologies and randomized controlled trial need to be performed to state such claims

Reviewer #2 (Remarks to the Author): with expertise in sarcoma therapy

In their manuscript, “Phase II Trial with Correlative Genomic Analysis of Durvalumab plus Pazopanib Combination in Patients with Advanced Soft Tissue Sarcomas,” Cho et al. report on a Phase II clinical trial combining pazopanib and durvalumab in patients with advanced soft tissue sarcomas. In this unselected population of patients with soft tissue sarcoma, they observed a 30.4% response rate and a median PFS of 7.7 months with expected toxicities. The authors performed an exploratory transcriptomic analysis and found a B lineage signature was associated with response to this treatment combination. Similarly, tumors with a high B cell infiltration and increased vessel density by immunohistochemistry responded better to pazopanib/durvalumab.

Overall, this is an important clinical trial. Although a similar combination of axitinib and pembrolizumab was previously shown to have similar efficacy (PMID: 31078463), the prior trial was driven by a high proportion of patients with ASPS. These results demonstrate that the combination of an anti-angiogenic TKIs with immune checkpoint inhibition has promising efficacy in a broader range of soft tissue sarcomas, and the authors have started to follow this up with a randomized trial.

The authors should be applauded for their correlative analyses, but there are several issues that need to be addressed prior to publication.

Major Points

1. The authors noted that no genomic alterations were associated with response to treatment, but it seems like CDKN2A alterations were much more common in non-responders (Figure 3A, 63% vs. 0%). How was this analysis performed, and why wasn't CDKN2A mutation associated with response?
2. The authors found that the B lineage signature, but not the T cell or CD8 T cell signatures, measured using MCP counter was higher in responders. However, CD3+ and CD8+ density appeared to be much more strongly associated with response than CD20+ density by immunofluorescence. How do the authors explain this discrepancy? Does the B lineage signature from MCP counter correlate with CD20+ density by IF? If B cells are the most important for response, can the authors confirm their importance using another deconvolution tool (CIBERSORT, TIMER, etc.)?
3. Given that there are more non-responders than responders, splitting at the median for the survival analysis in Figure 4 and Appendix Figure 7 doesn't make a lot of sense. This may explain why B cells seem to outperform CD8 T cells for stratifying patients by PFS. It would be less biased to consider the cell densities by IF as continuous variables and perform a Cox regression analysis.
4. The authors should provide more information on how patients were assigned to Sarcoma Immune Classes because the original paper by Petitprez et al (PMID: 31942077) was inconsistent across the cohorts analyzed. Did the authors use all of the MCP counter signatures except fibroblasts (as was used for samples from TCGA) or just T cells, cytotoxic lymphocytes, B lineage, and endothelial cells (as was used in the SARC028 cohort)? Petitprez et al empirically redefined their centroids for their SARC028 immunotherapy cohort. Which centroids were used for this analysis? Line 156 should be reworded: What do they mean “Upon application of other predictors of Sarcoma Immune Classes”?
5. Sarcoma immune class E was associated with a high prevalence of tertiary lymphoid structures (TLS), which are easier to measure (can be observed on H&E). TLS have been shown to enrich for patients who respond to immunotherapy (PMID: 35618839). Were TLS associated with response on this trial?

6. Line 234: “Given the dramatic response demonstrated in UPS, ASPS, and ANG, frontline treatment should be pursued with PD-L1 blockade and pazopanib in combination.” This makes it seem like the authors are suggesting that this combination should be used in the front-line setting now. Additional studies are needed before this becomes standard of care for these patients.

Minor Notes

1. It would be worth clarifying that Dose in Fig. 2D is referring to pazopanib dose.
2. The authors found no correlation between PFS or OS and PD-L1 score. Was there any difference in PD-L1 expression between responders and non-responders?
3. The authors didn't compare the pathways in the middle of Figure 3B with response to therapy. Were any of these (such as TLS) predictive?
4. What do the vertical splits between the patients in Figure 3B represent?
5. Why did the authors use a t test for Appendix Figure 3, but Wilcoxon rank-sum tests for Figure 3C? The data distribution looks quite similar.
6. The authors performed multiplex immunofluorescence, but they only analyzed each of the markers separately. Were double positive cells like PD1+/CD8+ or PD1+/CD20+ cells associated with response?
7. B cell infiltration is not a “predictive” biomarker because patients with more B cells appear to have better outcomes regardless of treatment.
8. I would recommend breaking up the Results section into sub-sections with sub-headings.
9. Line 143 needs to be reworded: “none of which were not correlated with response to treatment.”
10. It would be very helpful for the sarcoma research community to deposit the raw RNA-Sequencing data in a publicly available database.

Reviewer #3 (Remarks to the Author): with expertise in biostatistics, clinical trial study design

The statistical methods are not clearly described.

- (1) For example, Kaplan–Meier method and log-rank test are only mentioned in the figure legend rather than stated in the Method section.
- (2) The authors doesn't consider multivariable regression (multiple linear regression or Cox regression) to control for patients factors, such as age, ECOG and disease stage. They listed these variables in Table 1, but didn't consider them when comparing different groups.
- (3) they didn't use multiple test adjustment when they compare expression levels of immune-checkpoint genes. They didn't provide the reason.

Reviewer #1 (Remarks to the Author): with expertise in sarcoma (immuno)therapy

Response: Thank you for your positive and encouraging comments. We hope that our responses below adequately address your concerns.

1. Abstract

-Highlight total TRAE grade 1/2 and 3/4 rate

Response: Thank you for your valuable feedback. We have described total TRAE grade 1/2 and 3/4 rate in the abstract based on your recommendation. Please see the line line 11 page 3 in the abstract.

“The most common treatment-related adverse events of any grade included fatigue (20 [42.6%]), anorexia (17 [36.2%]), and diarrhea (17 [36.2%]). The common treatment-related adverse events of grades 3–4 included neutropenia (9 [19.1%]), elevated aspartate aminotransferase (7 [14.9%]), alanine aminotransferase (5 [10.6%]), and thrombocytopenia (4 [8.5%]).”

2. Introduction

-Based on the Geddis trial demonstrated that dox = gem/doce, and considered first line option as well. AIM has never demonstrated OS benefit and really reserved for pts w high tumor burden/symptoms. Gem doce ORR is actually 18%, not <10% as above. Would fix intro to reflect appropriate treatment strategies/data in STS

Response: Thank you for your valuable suggestion. We have clearly described and revised the treatment strategies in the introduction according to your recommendation (page 5, line 3).

“Single agent doxorubicin is the preferred fist-line treatment for advanced STS, whereas anthracycline-based combinations (doxorubicin or epirubicin with ifosfamide) can be considered to relieve symptoms with significant tumors. Although it did not show improved overall survival compared to that with doxorubicin monotherapy in the GeDDiS trial, the gemcitabine/docetaxel combination be considered as an acceptable alternative in patients in whom doxorubicin is contraindicated”

-2nd paragraph, should also highlight the nivo/sutent trial (immunosarc) JITC 2020 Broto et al.
- Overall helpful to include prior IO + TKI studies/outcomes

Response: Thank you for your valuable suggestion. We addressed and highlighted the prior IO + TKI trials including nivolumab/sunitinib trial according to your recommendation. Please see line 22 in page 5.

“Furthermore, the nivolumab/sunitinib combination had favourable activities, with 48% of the 6-month PFS rate and 24 months of OS. The toxicity profile was generally tolerable regardless of the high rate of dose interruption and toxicity compared to those of anti-PD-1 or sunitinib monotherapy. Based on increasing evidence, to confirm a better efficacy and toxicity profile, further investigation with a combination of VEGF and PD-1/PD-L1 blockade for various subtypes of STS is needed.”

3. Results

A) Rationale in comparing ang/asps cohorts vs other, was this a predefined endpt? Would caution this analysis

Response: Thank you for your balanced review of this topic. We have addressed these points making some changes to the text as your suggestion.

First of all, in our previous Cox analysis related to PFS with clinicopathologic factors, ASPS/ANG histology (P=0.03), CD20+ B cell infiltration (P<0.01), and vessel density (P=0.03) were associated with improved PFS in univariate Cox regression analysis. Finally, CD20+ B cell infiltration was identified as the only independent predictor for PFS via multivariate analysis. We apologize for not to provide details of this information previously, and we have included this information in our revised manuscript as appendix table 5.

In addition, for soft tissue sarcoma, because of heterogenous nature, subgroup analyses according to the histology is an important issue. In previous clinical trials with VEGF and IO combination, substantially various response rate was noticed based on the proportion of ASPS cases. Furthermore, indolent nature of ASPS also results in the different PFS and OS for many studies. Therefore, we cautiously performed subgroup analysis to provide further informative signal for clinicians. We hope that the revisions made the manuscript according to reviewers' suggestions will enhance the quality of the manuscript further.

Please see line 16 page 11 in result sections as follows.

“In previous studies, various response rates were noted based on the proportion of ASPS (36% in the axitinib/pembrolizumab and 7% in sunitinib/nivolumab). In addition, the indolent nature of ASPS may influence PFS and OS. Therefore, although it is not statistically significant in multivariate analyses, we cautiously investigated the subgroup efficacy to provide further informative signals for clinicians. The non-ASPS/ANG cohort in our study achieved significantly better response (ORR: 23.8% vs 9.5%) and PFS (7.0 vs 3 months) than those obtained with the axitinib–pembrolizumab combination. Consequently, the ASPS/ANG as well as non-ASPS/ANG cohorts in our study showed favourable efficacies. This finding should be considered for hypothesis-generating purposes.”

B) 4 pts that were SIC E, what is their histology and Comment which were in SIC E, but did not respond

Response: Thank you for your valuable suggestion. We have described in detail histologic information for 5 cases with SIC E group. Please see line 17 on page 9.

“Regarding the SIC E class (n=5), DSCRT (n=1) had CR, SS (n=2) and UPS (n=1) had PR, whereas LMS (n=1) had SD, respectively.”

4. Methods

-How was vessel density performed? Method section is very scant, unclear how analyses were actually performed; not clear how to reproduce

Response: Thank you for your valuable comments. We previously described in supplementary method for further information about multiplex immunohistochemistry and vessel density. We described in methods sections and added more comment. Please see line 25 page 17.

“Manual annotation was performed to identify tumor regions of interest in all slides stained with hematoxylin-eosin (H&E). The density of specific cell types expressing CD3, CD8, CD20, or PD-1 was quantified by counting cells per mm². For CD31, positive density was determined as the proportion of positive pixels (including weak, moderate, and strong) relative to the total pixels within the regions of interest. These analyses were conducted on baseline tumor

samples with accessible tumor material. Further information can be found in the supplementary materials.”

5. Conclusion

A) Dox is not the only first line, refer to comment above

Response: Thank you for your comment. Based on the reviewer’s comment, we described the various treatment options of first line therapy in line 3 page 5 and line 9 page 11.

“Single agent doxorubicin is the preferred first-line treatment for advanced STS, whereas anthracycline-based combinations (doxorubicin or epirubicin with ifosfamide) can be considered to relieve symptoms with significant tumors. Although it did not show improved overall survival compared to that with doxorubicin monotherapy in the GeDDiS trial, the gemcitabine/docetaxel combination be considered as an acceptable alternative in patients in whom doxorubicin is contraindicated”

“Particularly, beyond the first-line treatment, an objective response rate of <16% and PFS of 2–4 months are reported”

B) Would caution analysis w non-ASP/Ang vs other, there is no clear precedence to do this, instead just highlight the ORR, and which histologies stood out.

Response: Thank you for your valuable comments. From the beginning of analyses, we performed Cox regression analysis to predict better efficacy. In univariate and multivariate analyses with classified prognostic factor (clinical, pathologic and genetic factors), B cell was found to be significant independent factor for response as well as PFS. We apologize for not provide in this information in detail previously, and we have included this information in our revised manuscript as appendix table 5.

In previous studies, substantially various response rate was noticed based on the proportion of ASPS. Furthermore, indolent nature of ASPS also results in the different PFS and OS. Therefore, previous research with nivolumab/sunitinib (PMID: 33203665) also explained the proportion, efficacy of ASPS vs non-ASPS. In addition, pembrolizumab/axitinib combination (PMID: 31078463) study also described the response and PFS for ASPS and non-ASPS. Therefore, we cautiously performed subgroup analysis to provide further informative signal for clinicians. We hope that the revisions made the manuscript according to reviewers’ suggestions will enhance the quality of the manuscript further.

Therefore, for further understanding of reviewers as well as readers, we added and depicted as follows in results section. Please see line 7 page 10 and line 16 page 11.

“Finally, in the univariate Cox regression analysis including histology, SIC, clinical factors (age, gender, primary sites, and stage), immune and vessel densities, ASPS/ANG histology (P=0.03), high CD20+ B cell infiltration (P<0.01), and high vessel density (P=0.03) were associated with improved PFS. Of these, CD20+ B cell infiltration was identified as the only independent predictor of PFS in multivariate analysis (Appendix Table 5).”

“In previous studies, various response rates were noted based on the proportion of ASPS (36% in the axitinib/pembrolizumab and 7% in sunitinib/nivolumab)^{13,14}. In addition, the indolent nature of ASPS may influence PFS and OS. Therefore, although it is not statistically significant in multivariate analyses, we cautiously investigated the subgroup efficacy to provide further informative signals for clinicians. The non-ASPS/ANG cohort in our study achieved significantly better response (ORR: 23.8% vs 9.5%) and PFS (7.0 vs 3 months) than those obtained with the axitinib–pembrolizumab combination. Consequently, the ASPS/ANG as well

as non-ASPS/ANG cohorts in our study showed favourable efficacies. This finding should be considered for hypothesis-generating purposes.”

C) Biomarkers of response in sarcoma that have correlated w efficacy include: TIL, PD1 expression - this should be noted

Response: Thank you for your valuable comments. Based on reviewer’s comment, we realized the importance to summarize various kinds of biomarkers in this section. Therefore, we have added and rephrased the sentence per your recommendation. Please see line 19 page 12.

“Despite the promising efficacy, little is known about the mechanism underlying the action of ICI and angiogenesis inhibitors in combination. Only limited numbers of studies have demonstrated the prognostic importance of the immune microenvironment and druggable markers in sarcoma with heterogenous significance; Sorbye et al reported a significant association of CD20+ B peritumoral lymphocytes with disease free survival and PD-L1 or NK cells were also reported as significant prognostic factors with multiple STS. However, most of these studies were performed with immunologic endpoints and only increased baseline plasma angiogenic activity showed such a correlation with treatment efficacy. Regarding the tertiary lymphoid structure (TLS), only 3 cases were TLS+ in our study (PR, n=2; SD, n=1). Finally, for the ICI and angiogenesis combination, only two of 14 responders could be detected via TLS positivity. Therefore, considering the controversial issues needing further optimization originating from the heterogeneity (sample and tumor types), we focused on exploring clinically applicable and reliable biomarkers. In accordance with previous results, transcriptomic and multiplex IHC analyses in our cohort showed that B-cell lineage significantly correlated with PFS and response. Regarding the tumour vessel, a significant relationship between high vascular density or CD31 with better response or longer survival in patients treated with sunitinib were reported. Therefore, to elucidate further determinants for the effects of the ICI and angiogenesis inhibitor combination, we performed an exploratory analysis with immune cells as well as microvessel density in archival formalin-fixed paraffin-embedded tissue samples. Although microvessel density was only significant in univariate analysis, considering the importance of the immunomodulatory functions of the tumor vasculature, combining angiogenesis and immune-related biomarkers may be worth exploring. The correlation between B-cell infiltration or vessel density and response to the ICI and angiogenesis inhibitor combination should be validated in a clinical trial comprising a larger set of patients.”

B) Would explain significance of vessel density as this is novel and how pazopanib may contribute to this density (or not)

Response: Thank you for your valuable comments. There are a few data evaluating the relationship between vessel density and treatment efficacy of angiogenesis inhibitors. We described the findings of the literatures in discussion section. Please see line 9 page 13.

“Regarding the tumour vessel, a significant relationship between high vascular density or CD31 with better response or longer survival in patients treated with sunitinib were reported. Therefore, to elucidate further determinants for the effects of the ICI and angiogenesis inhibitor combination, we performed an exploratory analysis with immune cells as well as microvessel density in archival formalin-fixed paraffin-embedded tissue samples”

C) Difficult to conclude that durvalumab + pazopanone should be front line; there are <10 pts treated on this trial w these histologies and randomized controlled trial need to be performed to state such claims

Response: Thank you for your balanced review. We agreed your valuable suggestions

Though there are various different histologic subtypes, soft tissue sarcoma was treated as a whole part. However, recently, based on the molecular/drug development, more treatment options are available. As shown in NCCN guideline (which suggest pembrolizumab in combination with axitinib for ASPS), we tried to suggest active treatment strategy with molecular understanding of specific histologies. Therefore, we analyzed the molecular background using OPAL IHC and depicted that higher expression of immune and vessel densities in ASPS, ANG, UPS compared to low expression of DDLPS and LMS. Based on the molecular and clinical data in accordance with previous clinical trial (pembrolizumab and axitinib), our information may provide informative signal for many clinicians as well as researchers. Nevertheless, we definitely agree that subgroup analysis is not enough to conclude the hypothesis. Therefore, according to the concern of reviewer's comment, we toned down and added more cautious comments as follows. Please see line 24 page 13.

“Given the dramatic response as well as high expression of immune and vessel densities demonstrated in UPS, ASPS, and ANG (Appendix Fig 11), PD-L1 blockade and angiogenesis inhibitor combination can be considered as a preferred option. Although our results are promising, benefit as first-line treatment must be confirmed with larger randomized trials.”

Reviewer #2 (Remarks to the Author): with expertise in sarcoma therapy

In their manuscript, "Phase II Trial with Correlative Genomic Analysis of Durvalumab plus Pazopanib Combination in Patients with Advanced Soft Tissue Sarcomas," Cho et al. report on a Phase II clinical trial combining pazopanib and durvalumab in patients with advanced soft tissue sarcomas. In this unselected population of patients with soft tissue sarcoma, they observed a 30.4% response rate and a median PFS of 7.7 months with expected toxicities. The authors performed an exploratory transcriptomic analysis and found a B lineage signature was associated with response to this treatment combination. Similarly, tumors with a high B cell infiltration and increased vessel density by immunohistochemistry responded better to pazopanib/durvalumab.

Overall, this is an important clinical trial. Although a similar combination of axitinib and pembrolizumab was previously shown to have similar efficacy (PMID: 31078463), the prior trial was driven by a high proportion of patients with ASPS. These results demonstrate that the combination of an anti-angiogenic TKIs with immune checkpoint inhibition has promising efficacy in a broader range of soft tissue sarcomas, and the authors have started to follow this up with a randomized trial.

The authors should be applauded for their correlative analyses, but there are several issues that need to be addressed prior to publication.

Response: Thank you for your positive and encouraging comments. We hope that our responses below adequately address your concerns.

1. The authors noted that no genomic alterations were associated with response to treatment, but it seems like CDKN2A alterations were much more common in non-responders (Figure 3A, 63% vs. 0%). How was this analysis performed, and why wasn't CDKN2A mutation associated with response?

Response: Thank you for the valuable comments. As the reviewer pointed out, the alteration of *CDKN2A* was significantly associated with response to treatment if both heterozygous losses and homozygous deletions are included ($P = 0.0028$). However, when we identified genomic correlates of treatment responses, only homozygous deletions were considered for tumor suppressor genes to provide a high-level biological confidence. As a consequence, *CDKN2A* homozygous deletion was not statistically associated with the response to treatment ($P = 0.53$). We apologize not to provide this information in detail, and we have included this information in our revised manuscript in line 23 page 8.

"In addition, amplification and homozygous deletions of any genes were not associated with the response."

2. The authors found that the B lineage signature, but not the T cell or CD8 T cell signatures, measured using MCP counter was higher in responders. However, CD3+ and CD8+ density appeared to be much more strongly associated with response than CD20+ density by immunofluorescence. How do the authors explain this discrepancy? Does the B lineage signature from MCP counter correlate with CD20+ density by IF? If B cells are the most important for response, can the authors confirm their importance using another deconvolution tool (CIBERSORT, TIMER, etc.)?

Response: Thank you for your valuable comments.

Because we used fresh tumor tissues for RNA sequencing, 28 cases were available. Contrary, immunofluorescence was done in 39 cases with archival formalin-fixed paraffin-embedded tissue. Therefore, the discrepancy between MCP counter and immunofluorescence might have been affected by different sample size. In addition, as the reviewer suggested, we have confirmed the correlation between B lineage scores from MCP counter and CD20+ density by IF (**Appendix Figure 9**). ($P = 0.0087$, $R = 0.51$ by Pearson, $P = 0.010$, $\rho = 0.50$ by Spearman). We also analyzed using the deconvolution tool, CIBERSORT, but unfortunately all of cell markers were not statistically associated with responses including B and CD8+ T cells (**Figure R1**) although responders showed the higher values compared to non-responders for the signatures of B naive, plasma, and CD8+ T cells. This might be due to different defined cell compositions depending on tools.

Appendix Figure 9. A correlation plot between CD20+ density by IF and B lineage score from MCP counter.

Figure R1. Cellular composition estimated by CIBERSORT.

Therefore, we have added Appendix Figure 9 and revised the sentence according to reviewer's comments. Please see line 12 page 10.

"In addition, B lineage scores from the MCP-counter were significantly correlated with CD20+ density by immunofluorescence ($P = 0.0087$, $R = 0.51$ by Pearson, Appendix Fig 9)."

3. Given that there are more non-responders than responders, splitting at the median for the survival analysis in Figure 4 and Appendix Figure 7 doesn't make a lot of sense. This may explain why B cells seem to outperform CD8 T cells for stratifying patients by PFS. It would be less biased to consider the cell densities by IF as continuous variables and perform a Cox regression analysis.

Response: Thank you for your valuable comments. From the beginning of analyses, we performed Cox regression analysis including classical prognostic factors (SIC group, histology, gender, age etc). To perform the univariate Cox regression analysis in combined with categorical variables, we used dichotomized (high and low) parameters in accordance with previous studies (PMID 22375962, PMID 31942077, and PMID 24086736). In our univariate analysis, ASPS/ANG histology ($P=0.03$), CD20+ B cell infiltration ($P<0.01$), and vessel density ($P=0.03$) were associated with improved PFS whereas CD8 and CD3 cell were not. In multivariate model with classified prognostic factor, B cell was found to be significant independent factor.

Therefore, CD20+ B cell infiltration was the only significant independent factor both for

response and PFS. According to reviewer's comment, we added in detail in our revised manuscript. We apologize not to provide this information previously, and we have included this information in our revised manuscript as appendix table 5. Please see line 7 page 10.

“Finally, in the univariate Cox regression analysis including histology, SIC, clinical factors (age, gender, primary sites, and stage), immune and vessel densities, ASPS/ANG histology (P=0.03), high CD20+ B cell infiltration (P<0.01), and high vessel density (P=0.03) were associated with improved PFS. Of these, CD20+ B cell infiltration was identified as the only independent predictor of PFS in multivariate analysis (Appendix Table 5).”

4. The authors should provide more information on how patients were assigned to Sarcoma Immune Classes because the original paper by Petitprez et al (PMID: 31942077) was inconsistent across the cohorts analyzed. Did the authors use all of the MCP counter signatures except fibroblasts (as was used for samples from TCGA) or just T cells, cytotoxic lymphocytes, B lineage, and endothelial cells (as was used in the SARC028 cohort)? Petitprez et al empirically redefined their centroids for their SARC028 immunotherapy cohort. Which centroids were used for this analysis? Line 156 should be reworded: What do they mean “Upon application of other predictors of Sarcoma Immune Classes”?

Response: Thank you for the valuable comments. The previous study by Petitprez *et al.* have showed that SIC E tumors were associated with the highest response to pembrolizumab compared with other SIC classes, which is consistent with our analysis. To predict SIC class in our cohort, we have used all the signature except for fibroblasts according to the original paper by Petitprez *et al.* As the reviewer pointed out, Petitprez *et al.* used only T cells, cytotoxic lymphocytes, B lineage, and endothelial cells for SIC class prediction of their immunotherapy cohort, but gene expression profiling of the immunotherapy cohort was derived from FFPE samples and performed by Nanostring Technology not RNA-seq. We have performed RNA-seq to obtain gene expression profiling of our study cohort, therefore, we used all signature except fibroblast according to the original paper. In the revised manuscript, we have included this information. Please see line 9 page 9.

“Regarding the TME signatures and the mRNA expression levels of immune-checkpoint-related genes encoding PD1, PDL1, CTLA4, and LAG3, the differences between responders and non-responders were not significant (Fig 3C and Appendix Fig 4). Upon application of the Sarcoma Immune Classes (SIC) classification predicted by the signature scores of the microenvironment cell populations (MCP)-counter except fibroblast, a B lineage-high signature (a hallmark of an immune-high E class) correlated with an improved PFS, unlike class B (immune-low). Four of the five patients (80%) who fell into class E were also responsive to the treatment (Appendix Fig 5), consistent with the previous pembrolizumab study”

5. Sarcoma immune class E was associated with a high prevalence of tertiary lymphoid structures (TLS), which are easier to measure (can be observed on H&E). TLS have been shown to enrich for patients who respond to immunotherapy (PMID: 35618839). Were TLS associated with response on this trial?

Response: Thank you for your valuable suggestion. Previously, according to the published article (PMID: 35618839), we analyzed TLS as well as SIC. Among the 39 cases, only 3 were TLS positive tumors. Of those 3 cases, 2 were partial responder and 1 were stable disease. Therefore, with durvalumab and pazopanib combination, only 2 of 14 responders could be detected via TLS positivity. In addition, because of its heterogenous nature, tumor sites (for example, abdomen vs extremity) and histology, soft tissue sarcoma is one of the most complex disease entity. In addition, there are still controversial issues for the standardized interpretation methods due to the type of samples (surgical vs biopsy, primary tumor vs metastatic sites), we focused on exploring more clinically available and reliable assay. Therefore, considering the further need to optimize interpretation with heterogenous human cancers, we focused on exploring clinically applicable and reliable biomarker. We have added this in results and addressed our deep considerations in discussion section as follows. Please see line 2 page 13.

“Regarding the tertiary lymphoid structure (TLS), only 3 cases were TLS+ in our study (PR, n=2; SD, n=1). Finally, for the ICI and angiogenesis combination, only two of 14 responders could be detected via TLS positivity. Therefore, considering the controversial issues needing further optimization originating from the heterogeneity (sample and tumor types), we focused on exploring clinically applicable and reliable biomarkers. In accordance with previous results, transcriptomic and multiplex IHC analyses in our cohort showed that B-cell lineage significantly correlated with PFS and response.”

6. Line 234: “Given the dramatic response demonstrated in UPS, ASPS, and ANG, frontline treatment should be pursued with PD-L1 blockade and pazopanib in combination.” This makes it seem like the authors are suggesting that this combination should be used in the front-line setting now. Additional studies are needed before this becomes standard of care for these patients.

Response: Thank you for your valuable suggestion. We have addressed these points making changes to the text as per your suggestion.

Though there are various different histologic subtypes, soft tissue sarcoma was treated as a whole part. However, recently, based on the molecular/drug development, more treatment options are available. As shown in NCCN guideline (which suggest pembrolizumab in combination with axitinib for ASPS), we tried to suggest active treatment strategy with molecular understanding of specific histologies. Therefore, we analyzed the molecular background using OPAL IHC and depicted that higher expression of immune and vessel densities in ASPS, ANG, UPS compared to low expression of DDLPS and LMS. In addition, in Cox regression analysis with continuous variables, ASPS/ANG histology (P=0.03), CD20+ B cell infiltration (P<0.01), and vessel density (P=0.03) were associated with improved PFS in the univariate Cox regression analysis. Based on the molecular and clinical data in accordance with previous clinical trial (pembrolizumab and axitinib), our information may provide informative signal for many clinicians as well as researchers.

Nevertheless, we definitely agree that subgroup analysis is not enough to conclude the hypothesis. Therefore, according to the concern of reviewer’s comment, we toned down and added more cautious comments as follows. We hope that the revisions made the manuscript according to reviewers’ suggestions will help enhance the quality of the manuscript further.

Please see line 24 page 13.

“Given the dramatic response as well as high expression of immune and vessel densities demonstrated in UPS, ASPS, and ANG (Appendix Fig 11), PD-L1 blockade and angiogenesis inhibitor combination can be considered as a preferred option. Although our results are promising, benefit as first-line treatment must be confirmed with larger randomized trials.”

Minor Notes

1. It would be worth clarifying that Dose in Fig. 2D is referring to pazopanib dose.

Response: Thank you for your valuable feedback. We clarified and changed “dose” as “pazopanib dose” in figure 2D.

Figure 2D. Swimmer plot. Each lane represents a single patient’s data. X-axis represents the duration of treatment for each patient.

2. The authors found no correlation between PFS or OS and PD-L1 score. Was there any difference in PD-L1 expression between responders and non-responders?

Response: Thank you for your valuable suggestion. We also previously analyzed every

efficacy parameters such as PFS, OS, response (responder vs non-responder) and progression free rate (PFR <6 months vs PFR ≥ 6 months). However, neither response (P=0.58) nor PFR at 6 months (P=0.76) were statistically significant.

3. The authors didn't compare the pathways in the middle of Figure 3B with response to therapy. Were any of these (such as TLS) predictive?

Response: We appreciate the valuable comment. None of the signature scores in the middle was significant. We apologize not to provide this information previously though we have already performed the analyses from the beginning. Therefore, we have included this information in our revised manuscript as Appendix figure 4. We have included this result in the revised manuscript. Please see line 9 page 9.

“Regarding the TME signatures and the mRNA expression levels of immune-checkpoint-related genes encoding PD1, PDL1, CTLA4, and LAG3, the differences between responders and non-responders were not significant (Fig 3C and Appendix Fig 4).”

Appendix figure 4. Signature scores of TME compared between responders and non-responders

4. What do the vertical splits between the patients in Figure 3B represent?

Response: Thank you for your valuable comments. Vertical line indicates different response, consists of 1 case (CR), 8 cases (PR), 16 cases (SD) and 3 cases (PD). Based on your comment, we clarified the vertical line as histology (CR/PR/SD/PD). In our revised manuscript, we have included this information in Figure 3B and figure legend.

Figure 3B. Transcriptomic correlates of clinical response to durvalumab and pazopanib combination treatment

5. Why did the authors use t test for Appendix Figure 3, but Wilcoxon rank-sum tests for Figure 3C? The data distribution looks quite similar.

Response: We greatly appreciate the careful review. For appendix figure 3, we have analyzed both Wilcoxon rank-sum or t-test, respectively. Unlike others, only tumor mutation burden has a different outcome between Wilcoxon rank-sum and t test. Regarding t-test, it was not statistically significant (P=0.123) but Wilcoxon rank-sum test was marginally significant (P=0.0411, Fig R2) though the similar data distribution. Therefore, because conservative approach is needed from a statistical point of view, we reported the t-tests. In addition, we have summarized which statistical tests were used in Method and corresponding figures of the revised manuscript. Please see line 24 page 18.

Fig R2. Association between treatment responses and TMB (A) and neoantigen burden (B). P-values were determined via Wilcoxon rank-sum test.

“Statistical associations between continuous and categorical variables were evaluated using Wilcoxon rank-sum statistics. Genomic correlates of response to treatment were evaluated by Fisher’s exact test. Survival was plotted using Kaplan–Meier curves and compared using the log–rank test. In the case where other statistical tests were applied, we have specified which test was used in the corresponding figure legend.”

6. The authors performed multiplex immunofluorescence, but they only analyzed each of the markers separately. Were double positive cells like PD1+/CD8+ or PD1+/CD20+ cells associated with response?

Response: Thank you for your valuable comments. We previously analyzed the association of treatment efficacy (response and PFS) with double positive cells including PD1+/CD8+, PD1+/CD3+, and PD1+/CD20+ cells. However, none of those double positive cells (PD1+/CD8+ [P=0.395 for response, P=0.342 for PFS], PD1+/CD3+ [P=0.477 for response, P=0.361 for PFS], and PD1+/CD20+ [P=0.641 for response, P=0.192 for PFS]) were significant. Only CD20+ and vessel density were significant not only one-positive but also double-positive markers. Therefore, we suggested those as potentially relevant biomarkers for durvalumab and pazopanib combination.

7. B cell infiltration is not a “predictive” biomarker because patients with more B cells appear to have better outcomes regardless of treatment.

Response: Thank you for your valuable comments. We agree to the reviewer’s comment. Based on that, we edited as follows. Please see line 6 page 15.

“Our study also identified the potential role of B-cell infiltration as a valuable clinical decision-making tool in the prognostication of this heterogeneous tumour, although further investigations are needed.”

8. I would recommend breaking up the results section into sub-sections with sub-headings.

Response: Thank you for your valuable comments. We agree to the reviewer’s comment. Based on the reviewer’s suggestion, we broke up the results section into sub-sections with each sub-headings.

9. Line 143 needs to be reworded: “none of which were not correlated with response to

treatment.”

Response: According to the reviewer’s comment, we edited as follows “none of those were correlated with the response to treatment”

10. It would be very helpful for the sarcoma research community to deposit the raw RNA-Sequencing data in a publicly available database.

Response: We agree to reviewer’s suggestion. We will certainly cooperate to the sarcoma research community with the raw data. We provided this information in Data availability section.

Reviewer #3 (Remarks to the Author): with expertise in biostatistics, clinical trial study design. The statistical methods are not clearly described.

Response: Thank you for your positive and encouraging comments. We hope that our responses below adequately address your concerns.

(1) For example, Kaplan–Meier method and log-rank test are only mentioned in the figure legend rather than stated in the Method section.

Response: Thank you for your valuable feedback. The statistical description was added in methods section. Please see line 24 page 18.

“Statistical associations between continuous and categorical variables were evaluated using Wilcoxon rank-sum statistics. Genomic correlates of response to treatment were evaluated by Fisher’s exact test. Survival was plotted using Kaplan–Meier curves and compared using the log–rank test. In the case where other statistical tests were applied, we have specified which test was used in the corresponding figure legend.”

(2) The authors doesn't consider multivariable regression (multiple linear regression or Cox regression) to control for patients factors, such as age, ECOG and disease stage. They listed these variables in Table 1, but didn't consider them when comparing different groups.

Response: Thank you for your valuable comments. We already performed Cox regression analysis from the beginning. In our previous Cox analysis, ASPS/ANG histology ($P=0.03$), CD20+ B cell infiltration ($P<0.01$), and vessel density ($P=0.03$) were associated with improved PFS in the univariate Cox regression analysis. Finally, CD20+ B cell infiltration was identified as the only independent predictor for PFS via multivariate analysis. We apologize not to provide this information in detail previously, and we have included this information in our revised manuscript as appendix table 5. Please see line 7 page 10.

“Finally, in the univariate Cox regression analysis including histology, SIC, clinical factors (age, gender, primary sites, and stage), immune and vessel densities, ASPS/ANG histology ($P=0.03$), high CD20+ B cell infiltration ($P<0.01$), and high vessel density ($P=0.03$) were associated with improved PFS. Of these, CD20+ B cell infiltration was identified as the only independent predictor of PFS in multivariate analysis (Appendix Table 5).”

(3) they didn't use multiple test adjustment when they compare expression levels of immune-checkpoint genes. They didn't provide the reason.

Response: Thank you for the valuable comment. Since we have evaluated the statistical association for only two conditions (responders and non-responders), we did not perform multiple test adjustment. We deleted this sentence in our revised manuscript to avoid any confusion. Please see the legend of Figure 3B.

REVIEWER COMMENTS

Reviewer #1 (Remarks to the Author):

The edits improved the manuscript, however there are a few remaining issues

1- Gem/DOCE = doxorubicin, it isn't an alternative only when doxorubicin is contraindicated, pls correct that in the introduction

2- nivo/sutent was toxic to the point that only 25mg of sutent could be combined w nivo; pls modify the incorrect statement in the introduction on this matter

3- the value of combining ASPS/ang together vs other doesn't make any biologic sense; these are 2 diff diseases and there is no precedence to combine; would remove this

Reviewer #2 (Remarks to the Author):

I thank the authors for responding to my comments and suggestions. I think the clinical outcomes are the biggest strength of the study. The correlative findings will need to be confirmed in additional cohorts. I have a few small additional edits that I would recommend prior to publication:

1. It should be noted in the text that PD-L1 expression was not correlated with response.
2. It would be worth mentioning in the text that CDKN2A alterations were associated with response to treatment if you consider both heterozygous losses and homozygous deletions because there appears to be a strong effect.
3. The decision to use a Wilcoxon rank-sum test or t-test should be based on the distribution of the data not whether the p-value is significant. If the authors are not certain that their data is normally distributed, they should use the Wilcoxon rank-sum test throughout the manuscript.

Reviewer #3 (Remarks to the Author):

Reviewer #2 Minor #5.

The response on using a conservative approach (picking the insignificant p-value in t-test vs the significant p-value in Wilcoxon test) is not correct. The author should not select the test based on the test result. T-test did not show a significance maybe due to data not following a normal distribution. If the sample size is small, Wilcoxon test is preferred over t-test.

Reviewer #3 #3) they didn't use multiple test adjustment when they compare expression levels of immunecheckpoint genes.

The authors responded by saying that they did not perform multiple test adjustment because they have evaluated the statistical association for only two conditions (responders and non-responders). However, the multiple adjustment is due to multiple gene expression analyses (CD3+ T cell, CD8+ T cell and CD20+ B cell, not because you have multiple conditions.

Reviewer #1 (Remarks to the Author): Thank you for your positive and encouraging comments. We hope that our responses below adequately address your concerns.

The edits improved the manuscript, however there are a few remaining issues

1 Gem/doce = doxorubicin, it isn't an alternative only when doxorubicin is contraindicated, pls correct that in the introduction

Response: Thank you for your valuable suggestion. We have addressed these points making some changes to the text as your comment. Please see line 5 page 5

“Although it did not show improved overall survival compared to that with doxorubicin monotherapy, the gemcitabine/docetaxel combination can be considered based on the GeDDiS trial”

2 nivo/sutent was toxic to the point that only 25mg of sutent could be combined w nivo; pls modify the incorrect statement in the introduction on this matter

Response: Thank you for your valuable comment. We agree your valuable suggestions and revised as follows. Please see line 21 page 5.

“Furthermore, the nivolumab/sunitinib combination had favourable activities, with 48% of 6-month PFS rate and 24 months of OS, but they used lower dose of sunitinib (25 mg) due to high rate of dose interruption and toxicity”

3. the value of combining ASPS/ang together vs other doesn't make any biologic sense; these are 2 diff diseases and there is no precedence to combine; would remove this

Response: Thank you for your valuable comments. We removed all the comments with subgroup analysis combining ASPS/ANG in our manuscript.

Reviewer #2 (Remarks to the Author): Thank you for your positive and encouraging comments. We hope that our responses below adequately address your concerns.

I thank the authors for responding to my comments and suggestions. I think the clinical outcomes are the biggest strength of the study. The correlative findings will need to be confirmed in additional cohorts. I have a few small additional edits that I would recommend prior to publication:

1. It should be noted in the text that PD-L1 expression was not correlated with response.

Response: Thank you for your valuable suggestion. We noted the comment regarding the PD-L1 expression and response correlation as follows. Please see line 17 page 7.

“PD-L1 expression (combined positive score ≥ 1) was observed in 50% of participants (n = 23), and it was not correlated with responses (P=0.58).”

2. It would be worth mentioning in the text that CDKN2A alterations were associated with response to treatment if you consider both heterozygous losses and homozygous deletions because there appears to be a strong effect.

Response: Thank you for your valuable comment. We agreed your valuable suggestions and noted the comment as follows. Please see line 22 page 8.

“Although *CDKN2A* alterations combining both heterozygous and homozygous deletions were significantly associated with response (P = 0.0028), presence of homozygous deletion was not statistically significant (P = 0.53).”

3. The decision to use a Wilcoxon rank-sum test or t-test should be based on the distribution of the data not whether the p-value is significant. If the authors are not certain that their data is normally distributed, they should use the Wilcoxon rank-sum test throughout the manuscript.

Response: Thank you for your valuable comment. We agree to your comments and revised Appendix Figure 3 as Wilcoxon rank-sum test the same as other analysis. Therefore, all the statistical associations between continuous and categorical variables were evaluated using Wilcoxon rank-sum statistics throughout the manuscript and revised as follows.

“We evaluated the association between treatment responses and tumour mutation burden (TMB), neoantigen burden, and human leukocyte antigen (HLA) loss of heterozygosity (LOH), which have been reported to be potential predictive biomarkers for immunotherapy in other tumour types¹⁵⁻¹⁷, only TMB was statistically related to the treatment response in our sarcoma cohort (Appendix Fig 3).”

“While TMB status is not useful predictor to ICI in sarcoma because neoantigen levels are not correlated with CD8 T cells, ours demonstrated marginal statistical significance and further study may be needed to clarify this finding.”

Reviewer #3 (Remarks to the Author): Thank you for your positive and encouraging comments. We hope that our responses below adequately address your concerns.

Reviewer #2 Minor #5.

The response on using a conservative approach (picking the insignificant p-value in t-test vs the significant p-value in Wilcoxon test) is not correct. The author should not select the test based on the test result. T-test did not show a significance maybe due to data not following a normal distribution. If the sample size is small, Wilcoxon test is preferred over t-test.

Response: Thank you for your valuable comments. We revised Appendix Figure 3 as Wilcoxon rank-sum test the same as other analysis. Therefore, statistical associations between continuous and categorical variables were evaluated using Wilcoxon rank-sum statistics and revised as follows.

“We evaluated the association between treatment responses and tumour mutation burden (TMB), neoantigen burden, and human leukocyte antigen (HLA) loss of heterozygosity (LOH), which have been reported to be potential predictive biomarkers for immunotherapy in other tumour types¹⁵⁻¹⁷, only TMB was statistically related to the treatment response in our sarcoma cohort (Appendix Fig 3).”

““While TMB status is not useful predictor to ICI in sarcoma because neoantigen levels are not correlated with CD8 T cells²⁵, ours demonstrated marginal statistical significance and further study may be needed to clarify this finding.”

Reviewer #3 #3) they didn't use multiple test adjustment when they compare expression levels of immune checkpoint genes.

The authors responded by saying that they did not perform multiple test adjustment because they have evaluated the statistical association for only two conditions (responders and non-responders). However, the multiple adjustment is due to multiple gene expression analyses (CD3+ T cell, CD8+ T cell and CD20+ B cell, not because you have multiple conditions.

Response: Thank you for the valuable comments. As the reviewer suggested, we performed multiple test adjustment to evaluate the statistical differences between responders and non-responders in expression levels of immune checkpoint genes (Table R1). None of the immune-checkpoint genes shown in our manuscript demonstrated any statistical differences in their expression levels between responders and non-responder, both before and after p-value adjustment.

Gene	p-value	FDR-adjusted p-value
------	---------	----------------------

PDCD1 (PD1)	0.0550	0.606
CD274 (PDL1)	0.438	0.824
PDCD1LG2 (PDL2)	0.332	0.764
CTLA4	0.712	0.904
LAG3	1.00	1.00

Table R1. P-values from Wilcoxon rank-sum test between responders and non-responders for expression levels of immune-checkpoint genes.

In addition, the expression scores for cell lineage were calculated MCP-counter (R package) which provided 10 MCP estimate scores (T cells, CD8 T cells, Cytotoxic lymphocytes, B lineage, NK cells, Monocytic lineage, Myeloid dendritic cells, Neutrophils, Endothelial cells, and Fibroblasts) for each sample based on gene expression profiling. Therefore, we performed differential expression analysis (using DESeq2) between responders and non-responders and obtained the p-values and adjusted p-values for MCP-counter marker genes. Among eight marker genes, three were significantly up-regulated in responders compared with non-responders after p-value adjustment. B lineage markers showed the highest proportion of significantly up-regulated genes among 10 TME cell types (Figure R1). Table R2 summarized the MCP genes with p-value < 0.05.

Figure R1. The number of significant or non-significant marker genes after p-value adjustment

Gene	Cell	p-value	p.adj	Significance
CD3D	T cells	0.03471	0.30859	
ICOS	T cells	0.01806	0.22555	
TNFRSF25	T cells	0.00058	0.03380	
EOMES	Cytotoxic lymphocytes	0.03731	0.31806	

KLRD1	Cytotoxic lymphocytes	0.01868	0.22931	
CD19	B lineage	0.00000	0.00086	***
CD22	B lineage	0.00202	0.06764	
CD79A	B lineage	0.00005	0.00672	**
CR2	B lineage	0.02878	0.28223	
MS4A1	B lineage	0.00065	0.03559	*
CD160	NK cells	0.00730	0.13920	
CLEC10A	Myeloid dendritic cells	0.01583	0.21107	
TNFRSF10C	Neutrophils	0.00296	0.08425	
BMP6	Endothelial cells	0.00043	0.02896	*
BMX	Endothelial cells	0.00089	0.04213	*
EDN1	Endothelial cells	0.00067	0.03627	*
KDR	Endothelial cells	0.00699	0.13656	
MMRN1	Endothelial cells	0.03984	0.32865	
PALMD	Endothelial cells	0.01308	0.19162	
ROBO4	Endothelial cells	0.01420	0.19864	
SHANK3	Endothelial cells	0.02625	0.26972	
VWF	Endothelial cells	0.00230	0.07294	
PAMR1	Fibroblasts	0.00229	0.07263	
TAGLN	Fibroblasts	0.04408	0.34069	

Table R2. MCP genes with p-value < 0.05 via DESeq2

We noted this comment as follows in the manuscript. Please see line 7 page 9.

REVIEWERS' COMMENTS

Reviewer #3 (Remarks to the Author):

The authors have addressed all my concerns